# B Cells as an Immune-Regulatory Signature in Ovarian Cancer

**DOI:** 10.3390/cancers11070894

**Published:** 2019-06-26

**Authors:** Prachi Gupta, Changliang Chen, Pradeep Chaluvally-Raghavan, Sunila Pradeep

**Affiliations:** Department of Obstetrics and Gynecology, Medical College of Wisconsin, Milwaukee, WI 53226, USA

**Keywords:** ovarian cancer, B cells, tumor microenvironment, immune cells, tumor infiltrating lymphocytes

## Abstract

Increasing evidence suggests that the immune system plays a dynamic role in the progression of ovarian cancer, the deadliest gynecological malignancy worldwide. Accumulation of tumor-infiltrating lymphocytes has been associated with increased survival in ovarian cancer patients, and diverse interactions among immune cells in the tumor microenvironment determine tumor progression. While the regulatory functions of T cells among tumor-infiltrating lymphocytes are well defined and also involve therapeutic interventions, the role of B cells in ovarian cancer progression is still limited to their impact on survival. Recent studies have identified both pro- and anti-tumor responses of B cells in solid tumors, as different subsets of B cells play diverse roles in progression. Thus, in-depth characterization of B cell subtypes in each disease stage is crucial for understanding the importance and therapeutic potential of these cells in ovarian cancer. In this review, we summarize current knowledge about B cells in ovarian cancer and discuss emerging therapeutic interventions that could harness B cells to combat this deadly disease.

## 1. Introduction

Human cancers show divergent immunologic properties [1], requiring the immune system to continually adapt to tumor growth and to hone surveillance strategies [2]. To mediate effective tumor control, the immune system must recognize dynamic tumor heterogeneity and adopt new cycles of immune recognition and attack. Thus, understanding these mechanisms is crucial for developing immunotherapies that yield lasting responses.

Ovarian cancer, the most lethal gynecological malignancy in women worldwide, has the following subtypes [3]: endometrioid carcinoma, clear cell carcinoma, mucinous carcinoma, low-grade serous carcinoma and high-grade serous carcinoma (HGSC). Among these, HGSC accounts for ~68% of ovarian cancer and has the worst prognosis [3]. Regardless of advances in treatment, 70–80% of patients who initially respond to therapy ultimately relapse and die [4], often because the disease is diagnosed at late stages. However, accumulating evidence shows that the immunogenicity of ovarian cancer can open the door to immunotherapeutic approaches to treatment. For example, the presence of tumor-infiltrating lymphocytes (TILs) and their correlation with increased survival in ovarian cancer has validated the role of immunotherapy in ovarian cancer [5]. The identification of tumor-associated antigens (TAAs) in ovarian cancer also supports an immunotherapeutic treatment strategy [6].

The potential role of T cells in antitumor responses is well established and extensively studied. However, the contribution of B cells to tumor immune responses is less well understood. Apart from generating an antibody response against antigens, B cells can also interact with other immune cells through antigen presentation, cytokine secretion and expression of co-stimulating molecules [7]. In the tumor microenvironment, functionally distinct subsets of B cells are present, and the balance among subtypes may affect tumor development and behavior [7].

In this review, we highlight recent findings related to the contributions of B cells to pro- or anti-tumor responses in ovarian cancer and their potential relevance to ovarian cancer prevention.

## 2. B Cell Markers in Ovarian Cancer

B cell subsets—naïve B cells, memory B cells, plasma cells and regulatory B (Breg) cells—have been recognized in ovarian cancer. These subsets are identified by distinct molecular markers, as listed in Table 1. We did not include Bregs in the list, as they lack well-defined molecular markers in ovarian cancer, though different Breg phenotypes have been identified in mouse models and other cancer types [8].

**Table 1 cancers-11-00894-t001:** List of B cell markers used to characterize B cell subtypes in ovarian cancer.

Marker	Naïve B Cells	Memory B Cells	Plasma Cells
CD20	+	+	−
CD19	+	+	+
CD138	−	−	+
CD38	−/low	−/low	+
CD95	−	+	+
CD27	−	+	+
IGKC	−	−	+
IgG	−	+	+
IgD	+	−	−
IgM	+	+	−
CXCR5	+	+	−
CXCR3	−	+	+

Legend: The markers listed here have been used to study the prognostic significance of B cells in ovarian cancer [9,10,11,12]. Markers of Breg are not well defined in the literature: only IL-10 (Interleukin-10 (IL-10)) positive cells are being classified as Bregs [7].

## 3. Prognostic Role of B Cells in Ovarian Cancer

The prognostic significance of tumor-infiltrating lymphocytes has been widely recognized in cancer. For example, a systematic review by Maartje et al. [13] documented that, in most tumor types, B cells and plasma cells have a positive or neutral prognostic effect, with only a minority of studies reporting a negative effect.

In a study of HGSC, infiltration of CD19^+^ B cells in to the omentum was associated with poor survival [14]. Another study of metastatic ovarian carcinoma patients also showed that a higher percentage of CD19^+^ cells and natural killer (NK) cells predicted poor survival, supporting a role for B cells in ovarian cancer [15]. Contrary to those reports, CD20^+^ B-cells correlated with positive survival in a group of 199 ovarian cancer patients [16]. In a sample of 40 ovarian cancer patients, Nielsen et al. [9] demonstrated that CD20^+^ B cells co-localized with activated CD8^+^ TILs expressing antigen presentation markers and correlated with increased patient survival compared with CD8^+^ TILs alone. Santoiemma et al. [17] also showed that the presence of CD8^+^ and CD20^+^ TILs positively correlated with overall survival. Lundgren et al. [18] found that plasma cell infiltration in to epithelial ovarian cancer significantly impacted tumor progression and prognosis, as high CD138 expression correlated with significantly reduced overall survival. The discrepancies among these studies emphasize the need for more comprehensive studies that also focus on the interactions of B cells with other immune cells, and also clearly demonstrate the abundance of B cell subtypes in studied ovarian cancer patients.

## 4. Role of Regulatory B Cells (Bregs) in Ovarian Cancer

Bregs are a subset of B cells that produce IL-10 [19] but there are currently no precise surface markers for identifying them. B cells expressing surface molecular markers such as CD1 and CD5 have been shown to secrete IL-10, which is immunosuppressive. However, different Breg phenotypes have been described in different cancer types [19]. The immunoregulatory functions of Breg cells have been investigated in different cancer models, shifting our understanding of how Breg function is regulated during tumor progression has become frontier of cancer immunotherapy. Moreover, analysis of the expression of several immunosuppressive membrane-bound molecules (such as CD80, CD86, PDL1, PD1, Fas-L, CD40L and OX40L) and secretion of various cytokines (such as IL-10, TGF-β and IL-35) by Bregs in specific tumor settings and stages of cancer has revealed functional complexity among the subsets of tumor-evoked Bregs [20].

Various subsets of Bregs have been identified in diverse solid tumors such as ovarian, bladder, breast, pancreatic, colorectal, lung, squamous cells and hepatocellular [8], where they promote tumor growth and suppress cellular immune responses through diverse mechanisms. As demonstrated by a study in a 4T1 breast cancer murine model, a unique subset of Bregs (CD25^+^CD19^+^B220^+^ cells) constitutively expresses STAT3 and promotes tumor metastasis through TGFβ-dependent conversion of effector T cells to regulatory T cells (Tregs) [21].

Tumor-infiltrating B cells (TIL-B) acquire immunosuppressive properties as they come in to contact with tumor cells. Such tumor-evoked Bregs (tBregs) have been found in various human tumors [8]. Using an EMT-6 breast cancer model, Zhang et al. [22] found that TIL-B cells developed increased expression of PD-L1, CD86, CD80 and LAP/TGF-β in comparison to splenic B cells. These TIL-B cells also suppressed the proliferation of CD4^+^ and CD8^+^ T cells in vitro. Moreover, several factors that induce tBregs are found in the tumor microenvironment, including TNF-α (secreted by tumor cells) and IL-21 (secreted by T cells) [23,24]. Other than soluble factors, some of the cell–cell contact ligands, such as CD40L and PD-L1, that are expressed on tumor cells induce tBregs in the tumor microenvironment [7]. In addition to extensive studies on how the PD-1-PD-L1 axis allows tumors to escape from the immune system, a recent study showed that the eIF4F–STAT1–PD-L1 axis promotes PD-L1 expression on tumor cells and inhibitors of eIF4A (the RNA helicase component of eIF4F) could be developed as anticancer drugs, because inhibition of eIF4A downregulates PD-L1 expression on tumor cells [25].

Different signaling mechanisms have been identified in tumor-promoting Bregs. For example, one subtype of Bregs (CD19^±^CD1d^high^CD5^±^) promoted tumor growth in a mouse model of pancreatic ductal adenocarcinoma (PDAC) mediated by IL-35 signaling [26], making IL-35 a suitable therapeutic intervention in some human tumors. Bruton’s tyrosine kinase (BKT) signaling was also shown to suppress anti-tumor immunity by infiltrating lymphocytes in PDAC tumors. Activated BTK was identified in murine PDAC tumors, and was most prominent in CD19^+^ B cells and myeloid cells [27]. A BKT inhibitor, ibrutinib, is in clinical trials in various human tumors [8]

Breg cells that express GrB have also been seen in ovarian cancer tissue [18]. Moreover, Bregs induced the conversion of FoxP3^+^ Tregs from resting CD4^+^ T cells, supporting cancer metastasis [21]. A study in ovarian cancer patients showed an enriched population of IL-10^+^ B cells in ascites, and their frequencies positively correlated with Foxp3^+^ CD4^+^ T cells. Those B cells also suppressed IFN-g production by CD8^+^ T cells, showing that Breg cells can inhibit the antitumor adaptive immune responses in the tumor microenvironment [10]. Cancer cells could convert normal B cells to Bregs, thus inhibiting the antitumor immune response [19]. These findings have important clinical implications, as they suggest that Bregs must be controlled to interrupt the initiation of a key cancer-induced immunosuppressive event.

## 5. Humoral Immunity in Ovarian Cancer

Autoantibody responses associated with B cells have been found in many tumor types. Tumor-associated autoantibodies (AAbs) are produced as an immune response to aberrantly expressed, mutated, or post-translationally modified proteins or other autoantigens associated with tumors, and they may be enhanced by tumor-associated inflammation [28]. This class of biomarkers is highly valued, as AAbs exhibit several properties that make them attractive as early detection biomarkers. First, circulating antibodies are more stable over time than their corresponding antigens, which are subject to proteolysis, whereas antibodies are not [29]. Second, the immune response to tumor-associated antigens amplifies the signal. Therefore, AAbs may be detectable sooner than antigens themselves [29], potentially allowing detection at an earlier disease stage. Finally, AAb ELISA assays are readily translatable to clinical chemistry platforms. Therefore, AAb panels providing validated diagnostic discrimination could readily be added to the best available screening markers for clinical implementation.

Several AAbs have been identified in sera of ovarian cancer patients, and their levels correlate with specific tumor types and stages. A recent review by Fortner et al. identified 85 AAbs that discriminated between ovarian cancer cases and controls [30]. This study also analyzed a panel of AAb biomarkers to check their diagnostic performance in ovarian cancer. This panel, which included 11 AAbs (ICAM3, CTAG2, p53, STYXL1, PVR, POMC, NUDT11, TRIM39, UHMK1, KSR1 and NXF3), provided 45% sensitivity and 98% specificity for serous ovarian cancer. Single AAbs have much less sensitivity and specificity and therefore less potential as biomarkers. However, the clinical performance of any AAb biomarker depends highly on clinical stage, histology, age, treatment schedule and subjects’ general health. Thus, any biomarker study must keep all these factors in mind and associate clinically proven markers, such as CA125 and HE4 (in the case of ovarian cancer), with identified novel biomarkers. For example, a study by Wilson et al. [31] in early-stage HGSC, found that AAbs against HSF1 detected early stage malignancy better than CA125 alone and that combined measurements of anti-HSF1, anti-CCDC155 and CA125 might be useful for detecting early stage HGSC, thus highlighting the potential role of AAb biomarkers in detection of early disease. Another study, by Chatterjee et al. [32], suggested a role for AAbs in the surveillance of ovarian cancer and in predicting its recurrence, as AAbs to 5H6, HARS, CDR2 and Ro52 antigens predicted ovarian cancer recurrence 5.03 months before clinical onset in 21 patients, with a sensitivity of 90.5%. In contrast, levels of the known biomarker CA125 were below the standard cutoff (35 U/mL).

Most of the study considering the diagnostic potential of AAbs still have not discovered their role in prognosis of ovarian cancer. Although P53 AAbs have been associated with overall good survival in this cancer, there are still contradictory findings in case of the predictive value of p53 AAbs in ovarian cancer [33]. However, the prognostic utility of AAbs has been observed in other cancer types [34]. Also, an interesting finding about the therapeutic use of AAbs has been emerged. A circulating AAb against GRP78 protein was found to reduce invasiveness and increase apoptosis in ovarian cancer [35]. A follow-up study in chick embryo cultures of ovarian cancer cells treated the tumors with paclitaxel-loaded nanoparticles coated with anti-GRP78 antibodies [36]. That strategy decreased tumor growth compared with paclitaxel treatment alone, suggesting that delivering cancer drugs along with antibodies could be a powerful therapy against ovarian cancer. However, a recent study uncovered a pathogenic role for B cell-secreted antibodies in breast carcinoma: tumor-educated B cells and their derived antibodies formed a pre-metastatic niche, where they promoted lymph node metastasis by producing IgG that targeted HSPA4 [37]. Such studies warrant further investigations, as they showcase the importance of pathogenic antibodies in different tumor types.

## 6. Cross-Talk between B cells and the Tumor Microenvironment

The role of B cells extends beyond eliciting humoral immune responses. B cells can recognize antigens, regulate antigen processing and presentation, and modulate T cells and innate immune responses. Moreover, B cells can potentially influence all immune cells that express Fc receptors, such as dendritic cells (DC), granulocytes, natural killer cells (NK), and myeloid-derived suppressor cells (MDSCs). Also, studies of other cancers have suggested both pro-and anti-immune responses of B cells [7]. Due to the small number of investigations, however, the exact scenario of B cells in ovarian cancer is unknown.

A recent breast cancer study showed that B cells suppressed the antitumor immune response in tumor-bearing mice by expressing LAP/TGF-β and PD-L1, markedly reducing CD8^+^ T cell and CD49^+^ NK cell infiltration, and reducing cytolytic T cell response [22]. Another study that focused on the interaction between tumor cells and tumor-associated B cells in melanoma found that B cells induced a heterogenous population of cancer cells that phenotypically resembled cancer stem cells. Those B cells also induced therapy resistance by secreting IGF1 [38]. In contrast, Montfort et al. [11] found that B cell infiltration in to HGSC supported the development of an antitumor response. The strong memory B cell response was even enhanced by chemotherapy. The group also showed that omental B cells produced a network of cytokines and chemokines, including IFNg, IL12, GM-CSF and CXCL10, supporting an antitumor response. In line with these findings, another study showed that the presence of both CD8^+^ TIL CD20^+^ B cells in HGSC was highly prognostic, suggesting that cooperative interactions between these lymphocyte subsets strengthens antitumor immunity [12]. Thus, key factors that facilitate a coordinated CD8^+^ T cell and CD20^+^ B cell response should be defined, and immunotherapies should be designed to enhance not only CD8^+^ T cells but also CD20^+^ B cells in the fight against cancer [39]. However, B cells have been seen to foster squamous cell carcinoma (SCC) by activating Fcγ receptors (FcγRs) on resident and recruited myeloid cells and further repolarizing tumor-associated macrophages toward an immunosuppressive phenotype (M2-type) [40,41]. When PDAC-derived B cells were co-cultured with macrophages, PDAC-derived B cells also repolarized macrophages into the immunosuppressive and tumorigenic M2-type [27]. Such MDSCs acquire immunosuppressive and tumor-promoting functions, thus encouraging tumor growth and metastasis in tumor microenvironment [42,43].

B cells can achieve antitumor immunity by secreting IFNγ, facilitating CD4^+^ Th cells to polarize to Th1 responses, and promoting T cell expansion by presenting TAAs [7]. Conversely, Bregs were recently designated as immunosuppressive cells that can secrete anti-inflammatory mediators, such as IL-10, IL-35 and TGF-β, to convert T cells in to Tregs [44]. Recently, an excellent study by Ouyang et al. [45] showed that a subset of semi-mature dendritic cells (DCs) in the tumor microenvironment activated B cell differentiation toward the FcγRII low/IL-10 Breg phenotype. Another recent study, using IHC, found tumor-infiltrating CD20^+^ B cells in >50% of patients with HGSC exhibited a robust positive correlation with DC-LAMP^+^ DC density, in both the tumor stroma and tumor nests and also associated with the highest overall survival in HGSC [46]. Thus, this study highlights the roles of DC and B cell-mediated antitumor immunity in the HGSC tumor microenvironment.

## 7. B Cells Associate with Tertiary Lymphoid Structures (TLS) in Tumors

Tumor-associated TLS—that form at sites of inflammation—have been noted in cancer recently [47]. There is compelling evidences that immune responses can develop in TLS that are associated with tumor tissue independently of responses from secondary lymphoid organs [47]. The presence of TLS in tumor tissue has largely associated with a favorable prognosis for patients with solid tumor types such as lung, colorectal, breast and prostate, as well as in ovarian carcinoma [47].

Different types of immune cells, such as B cells, T cells and DCs, have been seen with in TLS. Also, the presence of B cells associated with TLS has correlated with better prognosis in non–small cell lung carcinoma (NSCLC) and colorectal, ovarian and pancreatic cancers [48,49,50,51]. More recently, in HGSC omentum tissue, the presence of B cells in TLS associated with the generation of a memory B cell response and with infiltration of DCs in to omentum tissue. Furthermore, TLS were shown to contain large aggregates of plasma cells that had a high level of IgG deposition [11]. Thus, Montfort et al. showed that B cells in TLS played a positive role in the antitumor response, though their presence did not correlate with patients’ survival. A study by Kroeger et al. [12] supported these findings in HGSC, as it reported high aggregates of antibody-producing plasma cells in TLS. The plasma cells correlated with a prognostically favorable CD8^+^ T cells response. Overall, the presence of B cells in TLS is a marker of increased patient survival in ovarian cancer and other solid tumor types.

## 8. Prospects and Conclusions

This review has focused on the field of B cell regulation in ovarian cancer. After a rigorous literature search, we realized that studies of B cells in ovarian cancer are very limited, though recent work with other cancer models has identified B cells as an attractive immunotherapeutic target. In the case of ovarian cancer, there could be a similar scenario, but there are still many unanswered questions. First, does the presence of B cells in the tumor vicinity support tumor growth or does it indicate anti-tumor activity? That role also needs to be associated with the subtype of B cell in the tumor microenvironment. As the presence of CD20^+^ B cells has been associated with increased survival in ovarian cancer patients, these cells may play an anti-tumor role. In contrast, the presence of Bregs imparts immunosuppressive effects, supporting tumor growth. Anti-CD20 antibodies targeted to B cells, such as Rituximab and obinatuzumab, have been developed, but they have not been very effective in several tumor models [8]. To augment the anti-tumor response, it will necessary to identify the key B cell subset that has a regulatory function.

The second emerging question in B cell biology involves immunosuppressive Bregs. Although Bregs have been detected in ovarian cancer and are known to be immunosuppressive, there are still no well-known surface markers for defining Bregs as prognostic biomarkers in that disease. It would also be interesting to know which cellular or soluble factors are involved in the induction of Bregs in cancer progression. Moreover, it is still unclear whether Bregs are specific subsets of B cells or whether all types of B cells can convert to immunosuppressive Bregs. Therefore, the exact mechanisms underlying the origin and differentiation of Bregs in the tumor microenvironment merit further investigation, as these factors reset the immune system.

The third interesting investigation would be to explore the role of antibody-producing plasma cells in ovarian cancer. Several AAbs have diagnostic potential in that disease, but it is unclear whether they attack tumor cells or play a tumor-supportive role. As shown by a few studies, AAbs can be used for targeted delivery of drugs into tumor cells. For example, another cell penetrating anti-DNA AAbs 3E10 has been isolated from alupus-prone murine model that can deliver proteins and drugs specifically to cancer cells [52]. Future investigations will explore this therapeutic intervention in ovarian cancer. However, the utility of AAbs seems to be limited by the immunosuppressive microenvironment and because of poor anti-tumor potency. Therefore, it would be interesting to find ways to enhance the potency of AAbs.

Finally, a considerable effort is needed to explain how B cells behave in the tumor microenvironment. As anti-B-cell treatments have shown limited efficacy in cancer patients, understanding immunosuppressive interactions among immune cells should lead to more patient-specific therapies. Another emerging goal would be to determine tumor cells’ capacity to regulate B cell proliferation and convert them into Bregs. The cytokines and cell–cell contact factors involved in these processes also need to be identified. Furthermore, how Bregs support the proliferation of Tregs and control the activation of other immune cells, such as NK cells, DC cells and myeloid-derived suppressor cells, should remain a field of active investigation (Figure 1).

Thus, this review highlights potential B cell-mediated therapies and showcases emerging areas of research in this field. As B cells play a central role in the tumor microenvironment, we anticipate that B cell-mediated immunotherapies will greatly benefit ovarian cancer patients.

## Figures and Tables

**Figure 1 cancers-11-00894-f001:**
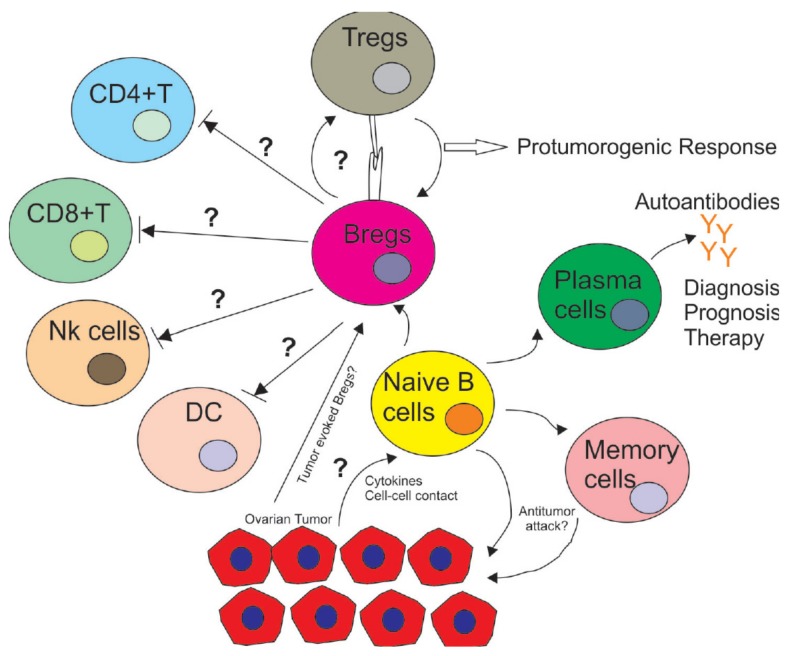
Immuno-regulatory role of B cells in the tumor microenvironment in ovarian cancer. The tumor microenvironment induces naïve B cells to differentiate into subsets, such as Bregs, plasma cells and memory B cells. These subsets play different roles as Bregs inhibit the proliferation of CD4^+^ T cells, CD8^+^ T cells, DCs and NK cells while supporting the proliferation of immune-inhibitory Tregs with unknown cytokines and cell–cell contacts. On the other hand, some subsets of B cells whose markers are unknown also induce an anti-tumor response with CD8^+^ TILs, inhibiting tumor progression. Plasma cells secrete autoantibodies that can play a variety of roles in cancer diagnosis, prognosis and therapy. It is unknown if autoantibodies can also support tumor development.

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
