# Peer review of "B Cells as an Immune-Regulatory Signature in Ovarian Cancer"

_cancers, 2019, doi:10.3390/cancers11070894_

Reviewer 1 Report

The review paper " B cells as an immune-regulatory signature in ovarian cancer" gives an acceptable overview on the current biological knowledge of B cells in ovarian cancer. This is not a systematic or comprehensive review. While this is well written, there are several issues that the authors need to clarify. At present, the study reads like a collaborative review. The authors mention a PUBMED search but it is unclear how studies were reviewed and assessed. Author should discussed in the text the information from Table 1.

Author Response

We appreciate the reviewer’s comments. For avoiding confusion, we have removed the PubMed search from the text. We have added the most recent research available on B cells in this review and have provided references for each study. Information of Table 1 has been added to the review.  

Reviewer 2 Report

The authors present a comprehensive and well written review on B cells in ovarian Cancer. The review has a straight structure and adresses several open questions by reviewing the available literature. 

I only have a few minor comments.

Introduction line 29: ovarian Cancer is not the most common gyn. Cancer. It is the most lethal. cervical Cancer is more frequent worldwide.

line 59: if studies showe positive or neutral prognostic effect, why does this suggest a collaborative effect with t-cells?

line 61: insert: was

line 76: Tumor markers for B-cells???

Paragraph 86 is repititive and redundant.

Author Response

The authors present a comprehensive and well written review on B cells in ovarian Cancer. The review has a straight structure and addresses several open questions by reviewing the available literature. I only have a few minor comments.

Response: We thank the reviewer for the positive comments.

Comment1: Introduction line 29: ovarian Cancer is not the most common gyn. Cancer. It is the most lethal. cervical Cancer is more frequent worldwide.

Response: As suggested by the reviewer ‘common’ has been replaced with ‘lethal’ in the revised review.

Comment 2: line 59: if studies showe positive or neutral prognostic effect, why does this suggest a collaborative effect with t-cells?

Response: We are thankful to the reviewer for pointing out this issue. We have removed this sentence from the review.

Comment 3: line 61: insert: was

Response: We inserted ‘was’ in the revised review.

Comment 4: line 76: Tumor markers for B-cells???

Response: It has been replaced by surface markers.

Comment 5: Paragraph 86 is repetitive and redundant

Response 5:  We have deleted the repetitive sentence from the text in the revised review.

Reviewer 3 Report

This manuscript systematically summarized recent studies which focused on the role of B cells in cancer progression, prognosis, and therapy. This is a well-organized and well-written paper about an important topic in cancer research and treatment. There are few reviews which focus on B cells in PubMed. I think the manuscript could provide useful information for the investigators who want to see the current studies about the role of B cells in cancer, especially for beginners.

Author Response

We thank the reviewer for the positive comments.